# Recent Advances in Transition Metal Tellurides (TMTs) and Phosphides (TMPs) for Hydrogen Evolution Electrocatalysis

**DOI:** 10.3390/membranes13010113

**Published:** 2023-01-15

**Authors:** Syed Shoaib Ahmad Shah, Naseem Ahmad Khan, Muhammad Imran, Muhammad Rashid, Muhammad Khurram Tufail, Aziz ur Rehman, Georgia Balkourani, Manzar Sohail, Tayyaba Najam, Panagiotis Tsiakaras

**Affiliations:** 1Department of Chemistry, School of Natural Sciences, National University of Sciences and Technology, Islamabad 44000, Pakistan; 2Institute of Chemistry, the Islamia University of Bahawalpur, Bahawalpur 63100, Pakistan; 3College of Physics, Qingdao University, Qingdao 266071, China; 4Laboratory of Alternative Energy Conversion Systems, Department of Mechanical Engineering, School of Engineering, University of Thessaly, Pedion Areos, 38834 Volos, Greece; 5Laboratory of Electrochemical Devices Based on Solid Oxide Proton Electrolytes, Institute of High Temperature Electrochemistry, RAS, 20 Akademicheskaya Str., Yekaterinburg 620990, Russia; 6Laboratory of Materials and Devices for Electrochemical Power Engineering, Institute of Chemical Engineering, Ural Federal University, 19 Mira Str., Yekaterinburg 620002, Russia

**Keywords:** hydrogen evolution reaction (HER), transition metal tellurides (TMTs), transition metal phosphides (TMPs), electrocatalysts, water splitting

## Abstract

The hydrogen evolution reaction (HER) is a developing and promising technology to deliver clean energy using renewable sources. Presently, electrocatalytic water (H_2_O) splitting is one of the low-cost, affordable, and reliable industrial-scale effective hydrogen (H_2_) production methods. Nevertheless, the most active platinum (Pt) metal-based catalysts for the HER are subject to high cost and substandard stability. Therefore, a highly efficient, low-cost, and stable HER electrocatalyst is urgently desired to substitute Pt-based catalysts. Due to their low cost, outstanding stability, low overpotential, strong electronic interactions, excellent conductivity, more active sites, and abundance, transition metal tellurides (TMTs) and transition metal phosphides (TMPs) have emerged as promising electrocatalysts. This brief review focuses on the progress made over the past decade in the use of TMTs and TMPs for efficient green hydrogen production. Combining experimental and theoretical results, a detailed summary of their development is described. This review article aspires to provide the state-of-the-art guidelines and strategies for the design and development of new highly performing electrocatalysts for the upcoming energy conversion and storage electrochemical technologies.

## 1. Introduction

Worldwide energy consumption has gradually increased due to the rise in population and the standard of living. Thus, renewable energy source usage has become increasingly important [1,2,3,4]. During water electrolysis for the production of hydrogen (H_2_), one of the most encouraging sustainable and clean energy carriers, only O_2_ is emitted as the byproduct without emissions of carbon [5]. Hydrogen has many excellent characteristics, including high energy density (140 MJ/kg) which is almost three times greater than typical fossil fuels (around 50 MJ/kg) [6]. Currently, the global annual production of H_2_ is about 500 billion cubic meters (bm^3^) [7,8]. The H_2_ produced is mainly used in industrial-scale applications such as petrochemical and petroleum-refining processes, fertilizers, chemical industries, and fuel cells [9,10,11,12]. H_2_ can be produced from various non-renewable and renewable energy resources, by employing different techniques, such as oil/naphtha reforming [13,14,15], methane-steam reforming [16,17,18], biomass [19,20,21], coal gasification [22,23,24], biological sources [25,26,27,28] and water electrolysis (WE) [29,30,31].

The different hydrogen production methods are comprehensively represented in Figure 1. Presently, 96% of the world’s H_2_ production derives from non-renewable sources such as fossil fuels, specifically from methane [32,33]. Furthermore, a lower quality of H_2_ and more harmful greenhouse gases are produced from fossil fuels [34,35,36].

Further, the limited fossil fuel reserves and the constantly increasing global energy demand require new energy methods with no carbon footprint. Hydrogen energy has recently gained attention due to its production from environmentally friendly methods such as electrocatalysis and photocatalysis, which can replace fossil fuel-based energy production [37]. Hydrogen can be produced using renewable resources such as water. Among the different H_2_-production approaches, high purity and eco-friendly H_2_ (99.999%) can be achieved from water electrolysis. The reaction is defined by Equation (1) [38],
2H_2_O + [Electricity 237.2 kJ/mol + Heat 48.6 kJ/mol] → 2H_2_ + O_2_
(1)

The efficiency of H_2_ production by H_2_O electrolysis is not sufficiently economically competitive, because of the high-energy utilization and the low-H_2_ evolution rate. Many researchers have been working on developing low-cost electrocatalysts to enhance performance and reduce energy consumption.

To boost the hydrogen evolution reaction (HER), transition metal (TM)-based catalysts are being increasingly used instead of the scarce and costly platinum group metals (PGMs). A variety of TM-related electrocatalysts, such as phosphides [39], nitrides, carbides [40], and chalcogenides [41], have been developed in recent years, with a similar HER performance to PGMs. Between these attractive catalysts, transition-metal tellurides (TMTs) and transition-metal phosphides (TMPs) have attracted much attention in recent years.

Compared with other chalcogens (oxygen (O), sulfur (S), and selenium(Se) [42] which may lead to higher electrical conductivity and enhanced covalent features, tellurium (Te) presents reduced electronegativity but higher metallic characteristics. The covalent feature can give a promising electronic-band structure by simplifying the edge arrangement of the conduction and valence bands with the H_2_O redox potential, and by boosting the TMT-center redox reactions. Additionally, TMT-based electrocatalysts own outstanding stability, abundant active sites, excellent conductivity, low overpotential, and strong electronic interactions between the components of the material, resulting in catalytic performance enhancement [43,44,45,46,47,48,49]. For example, Chevrel-phase Mo_6_X_8_ (X = Te, S, and Se) nanocatalysts for hydrogen evolution were studied [50], utilizing as HER-activity descriptor the Gibbs free energy of hydrogen adsorption (ΔG_H_). Interestingly, they identified that as the electronegativity of X-chalcogenide increases in Mo_6_X_8_, so does the hydrogen adsorption strength increase [50]. Lee et al. also confirmed that plenty of transition metal dichalcogenides (TMDs) with suitable anion-vacancy densities (e.g., MoTe_2_, TiTe_2_, and ZrTe_2_) are placed high on the volcano plot, facilitating the stronger hydrogen bond and thus promoting HER activity [51].

Transition metal phosphides (TMPs) have gained great scientific interest for their high corrosion resistance, good conductivity, and superior HER-electrocatalytic performance. The latest studies have shown that TMPs showed outstanding stability and activity in H_2_O electrolysis. Liu and coworkers [52] recently proved that CoNiP/Co_x_P/NF catalysts showed 290 mV overpotential at 10 mA cm^−2^ for HER with good long-term stability in natural water conditions.

Chang et al. [53] synthesized a catalyst (Fe, P-NiSe_2_ NFs) through chemical deposition at the gas phase, which displayed a huge current density at 1.8 V (800 mA cm^−2^) and constancy for more than 8 days. Moreover, Wu and colleagues [54,55] prepared the Ni_2_P-Fe_2_P catalysts that showed outstanding stability and activity, achieving current densities of 100 and 500 mA cm^−2^, with only 1.682 and 1.865 V required, in 1.0 M KOH-containing water, and the CoP_x_@FeOOH catalyst that exhibited good stability for 80 h at a current density of 500 mA cm^−2^ and an overpotential of 283 mV at 100 mA cm^−2^ in 1.0 M KOH-containing water.

There are examples of TMP- and TMT-based materials that showed outstanding performance related to HER activity in both acidic and alkaline media; some examples are mentioned here: Co_2_P@Cu nanostructure [56], Co_2_P/Ni_2_P/CNT [57], FeNiP/MoO_x_/NiMoO_4_/NF [58], Ni_2_P@NC/NF [59], MoP-Ru_2_P/NPC [60], CoTe_2_/Ti_3_C_2_T_x_ [61], NiFe_2_O_4_/NiTe [62], NiTe-HfTe_2_/g-C_3_N_4_ [63], Co,Ni-MoTe_2_ [64], and CoP/Ni_2_P@Co(OH)_2_ [65] (Table 1).

Considering the above, it can be concluded that TMPs and TMTs exhibited significant performance and stability for water electrolysis. This review article aims to provide a brief overview of the recent developments related to the various methods of water electrolysis for hydrogen production and the growth of transition metal telluride (TMT)- and transition metal phosphide (TMP)-based electrocatalysts for HER (Figure 2).

## 2. Water Electrolysis

The electrolysis of water (H_2_O) is an emerging method for H_2_ production because it utilizes renewable H_2_O, releasing only pure O_2_ as a side product. In addition, the electrolysis process uses DC power from renewable energy sources such as biomass, solar, and wind. Currently only 4% of H_2_ can be attained by electrolysis of H_2_O, mainly because of the high cost [66,67]. This value is expected to be improved with the improvement of renewable energy utilization (solar, wind, and nuclear). The European Energy Directive has achieved the goal of utilizing 14% renewable source-derived energy for the energy requirements up to 2020 [37]. H_2_O electrolysis has significant advantages, such as high hydrogen evolution rate, high purity, and good cell performance. The hydrogen purity is beneficial for its further transformation into electricity in low-temperature fuel cells (FCs) [29]. In an electrolysis procedure, the H_2_O molecule serves as the reactant, which under the effect of electricity dissociates into oxygen (O_2_) and hydrogen (H_2_). Water electrolysis, is divided into four types, depending on the operating conditions, electrolyte, and ionic agents (OH^−^, H^+^, and O^2−^). The usual water electrolysis approaches that will be analyzed, are; (i) alkaline water electrolysis (AWE), (ii) proton-exchange membrane water electrolysis (PEMWE), (iii) solid oxide electrolysis (SOE) [68,69,70,71,72], and (iv) microbial electrolysis cells (MEC) [73,74].

### 2.1. Alkaline Water Electrolysis (AWE)

Hydrogen can be synthesized from alkaline H_2_O electrolysis, which is a well-known approach up to the megawatt range; it is reported [75] that this phenomenon was invented in 1789 by Troostwijk and Diemann. In the alkaline H_2_O electrolysis procedure, at the cathode side, the alkaline solution of two molecules (KOH/NaOH) is reduced to a single molecule of H_2_ and two OH^−^ ions. The produced H_2_ is eliminated from the cathode surface, while OH^−^ is transferred from the cathode to anode through the porous diaphragm and converted into one molecule of water and a half (½) molecule of oxygen. The mechanism of alkaline water electrolysis is illustrated in Figure 3 (black color).

Alkaline electrolysis can occur at lower temperatures (30–80 °C), employing a KOH/NaOH aqueous solution as an electrolyte, whose concentration ranges from 30 to 20% [70,76,77,78]. In the alkaline H_2_O electrolysis procedure, nickel materials and asbestos diaphragms are used [72]. The diaphragm, positioned between the anode and the cathode is responsible for separating them, also separating the gaseous products that form at the corresponding electrodes, and preventing mixing of the produced gases during the electrolysis process.

However, there are some disadvantages concerning the alkaline water electrolysis process: low energy efficiency, lower current densities (below 400 mA cm^−2^), and low operating pressures [70,79]. The development of an anion exchange membrane (AEM), synthesized with anion-conducting polymers in the place of the asbestos diaphragm, is a novel approach in alkaline water electrolysis. This cutting-edge method seems exciting for H_2_O electrolysis in an alkaline medium [80,81,82,83,84].

### 2.2. Proton-Exchange Membrane Water Electrolysis (PEMWE)

The PEMWE was first introduced in 1959 by Grubb and the general electric company was established in 1966 to eliminate the problems associated with H_2_O electrolysis [85,86,87,88,89]. The PEMWE process is analogous to PEMFC, where the electrolytes used are solid polysulfonated membranes (e.g., Fumapem and Nafion) [90,91,92]. These membranes have various benefits such as higher proton conductivity (0.1 ± 0.02 S cm^−1^), lower gas permeability, higher-pressure operation ability, and lower thickness. Given the environmental impact and sustainability, PEMWE is one of the most suitable and favorable processes for hydrogen production. Furthermore, other outstanding characteristics of PEMWE are: high current densities (over 2 A cm^−2^), compact design, good efficiency, small footprint, quick response, low operation temperature (20–80 °C), releasing only oxygen as a byproduct [90,91,93,94,95]. Moreover, balancing PEMWE plants—associated with commercial applications—is relatively easy. Noble metal (Pt/Pd)-based up-to-date materials are standard for HER at the cathode for PEM electrolysis due to their higher performance [94,95,96]; however, these materials make the PEM electrolysis process more expensive as compared to alkaline water electrolysis. From the above discussion, it is clear that the cost of the electrocatalysts for PEMWE reduces its efficiency on an industrial scale. After that, substantial research has been completed to improve the production of PEMWE spare parts at a low cost, so this technique has become favorable for commercial application [97]. The schematic representation of the PEMWE operation is given in Figure 3 (red color).

### 2.3. Solid Oxide Electrolysis (SOE)

Solid oxide electrolysis (SOE) was invented in the 1980s by Donitz and Erdle [98]. SOE has received a lot of attention because it converts electrical energy into chemical energy, producing pure H_2_ with high performance [97,99]. It operates at high pressures and temperatures (around 500–850 °C), employing steamed H_2_O and oxygen anion (O^2−^) conductors [100]; its working principle is illustrated in Figure 3 (purple color). Currently, conducting materials such as ceramic proton conductors prepared and tested in SOFCs are used. The interest in ceramic proton conductors for SOEs is increasing because of their excellent properties, such as superior ionic conductivity and superior performance compared to O^2−^ conductors at 500–700 °C [37]. SOE technology has the advantage of working at higher temperature values than the common electrolysis types; however, some degradation and stability issues must be addressed before its use on a commercial scale [69,101,102,103].

### 2.4. Microbial Electrolysis (ME)

Microbial electrolysis cells (MECs) can produce hydrogen from wastewater containing organic matter and renewable biomass. The operating principle of an MEC is the reverse of microbial fuel cells (MFCs) [73]. The MEC procedure was first developed in the Netherlands (2005) by Wageningen and Penn State Universities [73,104]. In the MEC procedure, electrical energy is used to initiate hydrogen production from organic matter. Firstly, the substrate is oxidized on the anode side by microbes, and secondly, electrons, CO_2_, and protons are produced. As electrons are transferred to the cathode by an external circuit and the protons reach the cathode through the electrolyte, the electrons combine with protons and produce H_2_. The MEC operation principle is schematically shown in Figure 3 (green color). Using the MEC procedure, the specific resistance on the anode side is higher than on the cathode. Thus, increasing the input voltage by 0.2–1.0 V can regulate the initiation of the HER procedure at the cathode side. Therefore, MEC requires less external voltage when compared to other kinds of water electrolysis for H_2_ production [105]. However, MEC technology is still under development and has hurdles to overcome, such as the electrocatalytic electrode materials, the high internal resistance, the H_2_ production rate, and the complicated design that has to be simplified before its use on a large scale [106].

## 3. Transition Metal Tellurides (TMTs)-Based Electrocatalysts for HER

The electrocatalytic performance of bulk transition metal tellurides (TMTs) was found to be quite mediocre based on prior research [107,108,109]. To enhance the catalytic activity, nanostructuring has become an attractive approach, because it not only allows the availability of more active sites, but also aids in the mass transfer of gaseous products and electrolytes [109,110,111]. For example, MoTe_2_ nanosheets (NSs) synthesized through a liquid exfoliation strategy demonstrated a notably improved HER performance in 0.5 M H_2_SO_4_ instead of bulk MoTe_2_ [108]. In recent years a lot of attention has been paid to the growth of hollow NiTe_2_ nanotubes (NTs) [112], hierarchical CoTe_2_ nanowires (NWs) [113] and core–shell CoTe_2_@NC nanoparticles (NPs) [114]. Ananthara et al. [115] prepared nanostructured NiTe_2_ with two different morphologies, for example, nanoflakes (NFs) and nanowires (NWs) that were obtained by hydrothermally treating Ni foam with NaHTe and Te powders, respectively. The NiTe_2_ NWs presented excellent HER performance as compared to NiTe_2_ NFs in both alkaline and acidic environments and, temporarily, demonstrated values of Tafel slope similar to (for 0.5 M H_2_SO_4_) or lower (for 1 M KOH) than those of the Pt/C parameters, illuminating adequate reaction kinetics [115]. A large electrochemically available surface area (ECSA) and a high charge transfer capacity were responsible for the enhanced activity of NiTe_2_ NWs.

### 3.1. Nanostructuring

Nanostructuring can efficiently enhance the catalytic activity of exposed active sites by improving their efficiency. Zhuang et al. [109] observed that 1T-MoTe_2_ thin films prepared by chemical vapor deposition (CVD) exhibited good HER performance. However, after ion-beam etching, the 1T-MoTe_2_ films presented clearly improved HER activity, achieving 100 mA cm^−2^ current density at a Tafel slope value of 44 mV dec^−1^ in 0.5 M H_2_SO_4_ solution and an overpotential of 296 mV [109]. Furthermore, the ion-beam etched 1T-MoTe_2_ films showed the improved stability of the catalytic operation, maintaining 87% of its initial current density, in contrast to the 40% achieved for the pristine sample after 3600 s of successive electrolysis [109]. The improvement was attributable to the highly exposed edges of the active sites, which were determined through conductivity measurements, density functional theory (DFT), and visualized copper electrodeposition calculation. However, as the investigators stated, the method needs to be improved to achieve the enhancement of the active sites on different materials, as it is still time-consuming and commercially unfeasible for the mass production of electrocatalysts [109].

Metal-organic frameworks (MOFs) are broadly utilized to synthesize catalysts, due to their (i) large specific surface area, (ii) spatially ordered microstructure, and (iii) high nanoporosity. Wang and coworkers [116] developed an encapsulated CoTe_2_ NPs composite in N-doped carbon nanotube frameworks, denoted as CoTe_2_@NCNTFs, by employing a template of ZIF (zeolitic imidazolate framework)-67 [116]. This MOF-derived electrocatalyst had open channels for efficient gas discharge, high conductivity, and a larger surface area; furthermore, it allowed enhanced electron transport, resulting in a lower Tafel slope value than the bulk CoTe_2_. Subsequently, CoTe_2_@NCNTFs exhibited excellent catalytic activity, requiring an overpotential of 208 mV to attain 10 mA cm^−2^ for HER in a 1.0 M KOH solution. The CoTe_2_@NCNTF provided 10 mA cm^−2^ when used as a bi-functional electrocatalyst for water splitting at a cell voltage of 1.67 V [116]. Wang et al. further explained that in MOF-derived nanostructures, the composition of cobalt telluride can be easily included by employing a similar technique [117]. The attained optimum Co_1.11_Te_2_/C electrocatalyst exhibited a high surface distribution of Co-ions and more reducible Co species than CoTe_2_/C and CoTe/C, which contributed to HER activity improvement (178 mV@10 mA cm^−2^ in 1 M KOH). The higher performance of Co_1.11_Te_2_/C was elucidated through DFT calculations, illustrating a perfect Gibbs free energy, as seen in Figure 4a.

### 3.2. In Situ Development

The in situ development of the electrocatalysts on the surface of electrodes has attracted considerable interest in recent years. According to the literature, such a structure can prevent the agglomeration and collapse of nanostructured electrocatalysts, enhancing the long-term durability of the electrodes [121,122,123,124]. In addition, the binder-free adhesion of the electrocatalysts to the current collector can lead to excellent mass diffusion and charge transfer. In this regard, several TMT-related HER self-supported electrodes were recently reported, such as FeTe_x_ NSs on Fe foam [125], CoTe_2_ NPs on Co foam [126], NiTe_2_ NWs on Ni foam [115], Cu_7_Te_4_ arrays on Cu foil [127], an NiTe_2_ NS array anchored on Ti mesh [110], and a CoTe_2_ NW array on Ti mesh [113], which will not be systematically described in this review study. TMTs will ultimately be changed into the corresponding metal oxy-hydroxide, because the Te species of the surface tend to be soluble and are the actual catalytically active components, as mentioned previously [128] and confirmed by Yang and co-workers, who developed the CoTe nanoarrays on Ni foam [129].

### 3.3. Nanostructure Engineering

Nanostructure engineering can improve electrocatalytic performance by tuning the morphology and structure to provide more catalytic active sites. The latter could locally modify the coordination conditions and chemical characteristics, improving the electrocatalytic performance by the ensemble effects or ligands. Doping with heteroatoms to control the electronic structure is now widely accepted, and various nonmetals such as S, and P [129,130,131] and transition metals including the Co, Ni, and Fe [129,130,131,132,133] elements have already been used to dope TMTs, boosting their catalytic activity.

He et al. [134], confirmed that Fe-doping into Mo/Te nanorods (NRs) could significantly enhance the catalytic stability. Fe-doping into Mo/Te NRs promoted the formation of a Mo moiety with a higher valence state, fostering a significant modification in the electronic state. Moreover, after doping with Fe, the Tafel slope value and the resistance of charge transfer through Fe-Mo/Te were decreased, while the analysis of the Tafel slope provided information that the main kinetic roots include a mixed step of MOOH or MO synthesis [134]. Additionally, doping with Fe has been shown to increase the Co_1.11_Te_2_@NCNTF catalytic activity. He and colleagues [118] developed the Fe-Co_1.11_Te_2_@NCNTF catalyst, obtained through tellurization (under Ar/H_2_ atmosphere) of Fe-etched ZIF-67 gas, which presented a blue shift in the XPS spectrum of the binding energy of Co2p, comparative to the pristine Co_1.11_Te_2_@NCNTF, as shown in Figure 4b, resulting in reduced electron density Co atoms and also higher peaks (concentration) of Co^3+^, which can enhance OER performance. This accurately characterized the excellent HER activity of Fe-Co_1.11_Te_2_@NCNTF which offered TOF values 10-fold higher than the un-doped Co_1.11_Te_2_ [118]. In contrast, Pan et al. showed that doped Mn into 1T-VTe_2_ facilitated the stabilization of the 1T-phase and developed a nanosheet-like morphology with high porosity and an enhanced surface area, thus improving the HER activity, compared to that of the un-doped 1T-VTe_2_ [135]. Furthermore, it is reported that the activity could be enhanced by hybridizing Ni nanoclusters (NiNCs) with Mn-doped 1T-VTe_2_ NSs (denoted as NiNCs-1T-Mn-VTe_2_ NS), compared to all other electrocatalysts studied.

### 3.4. Dopants

Dopants can affect electrocatalytic activity [136]. This was also proved in TMT-related electrocatalysts. According to DFT analysis, Gao and coworkers [64] observed that the simultaneous doping of Ni and Co into MoTe_2_ can effectively activate the phase transition of 2H to 1T, in contrast with the mono-atom doping. The experiments demonstrated that the Ni/Co co-doped MoTe_2_ markedly improved HER activity. Apart from anion doping, the cation combination can also efficiently boost the electrocatalyst performance. Wang and his partners [137] described that the doping with S changed the 2H-MoTe_2_ from an inactive to an active electrocatalyst, due to the modification of the electronic structure, whereby electrons accumulated on the surface of S atoms, acting like active sites, adsorbed more rapidly the intermediate (H*), boosting the HER performance. Moreover, mixed anion-doped TMTs and tellurides, such as freestanding CoNiTe_2_ NSs [138], Ni_1−x_Fe_x_Te_2_ hierarchical nanoflake arrays [139], MoSe_0.12_Te_1.79_ and MoS_x_Te_y_/Gr [140] solid solutions [141] were recently discovered as HER electrocatalysts. The majority of these materials suggest that the stoichiometry between two chalcogens or metals is the main parameter for modulating the intrinsic electrocatalytic performance by the electronic structure. Besides the composition and nanostructure engineering, hybridization, or heterostructuring of the active electrocatalyst with other active components have been attractive strategies for boosting the electrocatalytic performance. Particularly, such hybridization/heterostructuring can reveal a multitude of interfaces, permitting electronic-structure engineering and enhancing the electrocatalysts’ reactivity and selectivity. Furthermore, the collaborative consequence might appear in heterogeneous exposed interfaces resulting from the migration from one portion of the adsorbed reaction moieties to the other. Thus, there is an unprecedented unblocking of electrocatalytic reaction radicals and an enhancement of the overall reaction rate [142,143].

### 3.5. Heterostructure

A number of TMT-related heterostructured electrocatalysts have been freshly reported to illustrate enhanced HER activity, such as TMT nanostructures composited with a secondary TMT [144,145,146], a chalcogenide [119,147], an oxide/hydroxide [148,149,150], or a phosphide [151,152]. For example, Xu et al. [153] constructed Ni_3_Te_2_-CoTe hybrids, which were developed in a single-step on carbon cloth by a hydrothermal procedure, and their electrocatalytic activity was compared to every individual component (Ni_3_Te_2_ and CoTe). Ni_3_Te_2_-CoTe hybrids delivered a current density of 100 mA cm^−2^ at 392 mV overpotential, with a smaller Tafel slope value of 68 mV dec^−1^. Based on these findings, the hydroxyl group chemisorption into the electrocatalytic surface was a defined rate-measuring step. The investigators suggested that the enhanced activity stemmed from the incorporation of CoTe into the electrocatalyst, exposing more Ni_3_Te_2_ active sites, also confirmed by the double-layer capacitance and the denser states close to the Fermi level, using DFT analysis.

Xue et al. [119] presented a heterojunction composed of NiTe/NiS, resulting from NiS nanodots (NDs) coupling with NiTe nanoarrays using an ion-exchange method, as schematically depicted in Figure 4c. HRTEM micrographs revealed that the NiS NDs decorated the NiTe surface with high density (Figure 4d,e). The NiTe/NiS nanointerfaces underwent a significant alteration of their electronic structure, thus modifying the binding energy of the *OOH intermediates. This is justified by the influence of the ligand in the NiS/NiTe, since the Ni d-band center shifted to a lower-energy level compared to NiTe, as a result of the Ni to S electron transfer, reducing the intermediates’ binding strength on the electrocatalytic surface, and lowering in this way the barrier for the reaction. Subsequently, the hybrid electrocatalyst required an overpotential of 257 mV for 100 mA cm^−2^ and exhibited a Tafel slope of 49 mV dec^−1^ in 1.0 M KOH, which was significantly less than that of pure NiS and pure NiTe. In addition, the electrocatalyst displayed over 50 h stability at 50 mA cm^−2^, and a potential boost of approximately 6%.

Instead, Sun et al. [154] successfully developed NiFe and RuO_2_-layered double-hydroxides (NiFe-LDH and RuO_2_-LDH) on NiTe NR surfaces, synthesizing NiTe@NiFe-LDH and Ni-Te@RuO_2_ heterostructures, which served, respectively, as anodic and cathodic electrocatalysts for water splitting. At 1.63 V, the fabricated device delivered a current density of 200 mA cm^−2^ and was powered by a 1.5 V solar cell for the continuous electrolysis of water [154]. This outcome is comparable to the one presented by the same research group [155]. They employed Pt/C||NiTe@FeOOH electrodes in a two-electrode electrolysis cell. Pt/C||NiTe@FeOOH exhibited a voltage of around 1.7 V for the same current density (200 mA cm^−2^).

Another research group [120] recently designed heterostructured and dual-phase CoPeCoTe_2_ NWs with many interfaces that showed excellent HER activity in alkaline/acidic solutions. CoP-CoTe_2_ NWs was employed as a bi-functional electrocatalyst (HER and OER) in a bipolar water electrolysis membrane (BPWEM). The use of BPWEM permits HER to occur simultaneously in kinetically favorable acidic and alkaline solutions. In the ‘forward bias’ operation, the anion-exchange membrane (AEM) is at the cathode, while the cation-exchange membrane (CEM) is on the anode side. The electrochemical neutralization between H^+^ and OH^−^ ions occurs when these ions cross the bipolar membrane and recombine to form water molecules. This can facilitate the electrolysis of H_2_O by reducing the external electrical energy requirements. Using the CoP-CoTe_2_ electrocatalyst in a BPWEM could transport 10 mA cm^−2^ at 1.01 V cell voltage (in the ‘forward bias’ operation), and it provided 100 h of stable operation without prominent degradation, superior to that of the anion-exchange H_2_O electrolysis membrane (AEWEM), using the same electrode pair, CoP-CoTe_2_. Furthermore, CoP-CoTe_2_ exhibited good stability, thus improving the catalytic performance. The results of the catalytic performance are mentioned in Figure 4f. The ‘forward bias’ of the bipolar water electrolysis membrane (BPWEM) can be considered a promising candidate to replace the traditional proton-exchange water electrolysis membrane (PEWEM) and anion-exchange water electrolysis membrane (AEWEM) approaches, as it permits H_2_ evolution with low electrical energy consumption.

### 3.6. Nanocomposites

The synthesized Te/FeNiOOH-NCs and FeNiOOH-NCs catalysts were characterized with SEM [150]. The FeNiOOH-NCs showed a uniform nanocubic shape, as illustrated in Figure 5a_1_,a_2_. Furthermore, after the tellurization, the material retained its cubic shape (Figure 5b_1_,b_2_).

The Te/FeNiOOH-NCs catalyst showed better electrochemical results as compared to the FeNiOOH-NCs due to following reasons: (1) the surface area increased after tellurization, retaining the nanocubic structure, so improving the number of active sites for electrocatalysis. In addition, the Te-tailored nanocubic structure may have enriched edge active centers [156]. (2) The Fe sites in FeNiOOH-NCs were generated on the surface through Te metal due to in situ hydrothermal self-templating formation that likely boosts the FeOOH electrochemical activity. Furthermore, the TEM results also verified the nanocubic structure of the Te/FeNiOOH-NCs catalyst (Figure 5c). These results confirm that all edge regions are enriched with the components (Te, Ni, and Fe) of the material. Moreover, this is supported from the literature that it is the surface active centers that are more abundant in retailored FeNiOOH nanocube edges. STEM with EDX of the catalyst (Te/FeNiOOH-NC) mapping showed the existence of the Te, Fe, and Ni elements (Figure 5d) [150,156].

Furthermore, the synthesized FeNiOOH-NC and Te/FeNiOOH-NC nanocomposites were tested for HER in 1.0 M KOH. As depicted in the linear sweep voltammograms displayed in Figure 6a, an overpotential of 167 mV vs. RHE is required to deliver a current density of 10 mA cm^−2^ for the HER process. This overpotential is better than the commercially used Pt/C electrocatalyst (199 mV vs. RHE) or pure FeNiOOH-NCs (279 mV vs. RHE), suggesting the excellent HER performance of the Te/FeNiOOH-NC electrode [150].

Figure 6b illustrates the corresponding Tafel plots, in which Te/FeNiOOH-NCs display a lower Tafel slope (93 mV dec^−1^) than the commercial Pt/C (20%) (108 mV dec^−1^). The durability of the electrocatalyst (Te/FeNiOOH-NCs) for HER was also explored by applying 1000 cycles of CV scans. As seen from the LSV curves of Figure 6c, after 1000 CV cycles only a loss of 1% was observed in the current density on the Te/FeNiOOH-NCs electrode for HER [150]. Moreover, CP measurements for checking the stability of the electrocatalyst for HER were completed. Only a 0.5% loss at 20 mA cm^−2^ current density was reported for the potential after 10 h of operation, as illustrated in Figure 6d [150].

Post-characterizations (XRD and FESEM images [150]) revealed that the Te/FeNiOOH-NCs electrocatalyst maintained the same cubic shape morphology with no obvious alkali erosion but with slight agglomeration in only a few places. This minor potential loss which is probably attributed to the electrocatalyst’s hour-long interaction with the electrolyte during the long-term stability measurement can be justified [147,157]. Nevertheless, more detailed postcharacterization protocols must be established, providing a more profound knowledge of the catalyst’s post-HER morphology.

## 4. Transition Metal Phosphide (TMP)-Based Electrocatalysts for (HER)

One of the primary design challenges is to improve the corrosion resistance and selective oxidation of TMPs. As a result of the corrosion phenomenon, the electrocatalyst may be affected, and hypochlorite/Cl_2_ may be created; consequently, efficiency will fast decline. Investigators, to address these issues, have synthesized an attractive HER electrocatalyst—by rational design in structure and composition—that can directly electrolyze H_2_O. Currently, H_2_O splitting normally employs three types of electrolysis: (i) an alkaline solution and seawater, (ii) an alkaline solution with NaCl (which replaces the seawater), (iii) and seawater.

TMPs present outstanding activity toward H_2_O electrolysis, and the activity of the added electrolyte (in the alkaline solution) is significantly higher than in the H_2_O solution. To enhance the electrolysis performance of H_2_O, an interesting approach would be to alter the active site and electronic structure by utilizing heteroatoms (dopants or promoters), facilitating the interaction of the bulk solution and electrocatalyst surface. Thus, gas adsorption and desorption can be accelerated, electrolyte diffusion can be controlled, and electrocatalysis can be improved.

### 4.1. Dopant

Chang et al. [53] prepared P and Fe co-doped NiSe_2_ nanoporous films following a procedure consisting of three steps, namely, anodic treatment, chemical-vapor deposition, and electrodeposition. The as-prepared nanocatalyst displayed outstanding electrocatalytic performance. Figure 7a exhibits the simulative image of the crystalline structure for NFs of Fe, P-NiSe_2_. The four elements (P, Ni, Fe, and Se) coexist, presenting homogeneous distributions, as observed in Figure 7b, c.

Overpotentials of about 120 and 180 mV were required to provide current densities of 100 and 500 mA cm^−2^, respectively, in a 0.5 M KOH aqueous solution, with 90% infrared correction. The NFs of the Fe, P-NiSe_2_ electrocatalyst demonstrated superior activity in the seawater electrolyte compared to that of non-PGM electrocatalysts. At the same time, the electrocatalyst delivered about 800 mA cm^−2^ with a voltage of 1.8 V and maintained stable operation for 200 h in an electrolytic cell with a seawater electrolyte.

The Fe-doping provided the main active sites for the HER procedure, while P-doping formed a passivation layer that maintained the phosphorus–oxygen bonds, enhancing in this way the electronic conductivity and hindering Se dissolution. Moreover, bi-metal phosphides enhance electrocatalytic activity and long-term stability, by the simultaneous influence of the different metals. Thus, the development of an ion-exchange phosphorylation electrocatalyst, in situ into a nickel-foam (NF) substrate, was discussed, that could efficiently alter the distribution of electrons locally for the central metal, thus promoting the synthesis of low-cost and highly active seawater-electrolytic catalysts.

### 4.2. Two-Dimensional Nanosheet Structure

Wu et al. [55] synthesized a Ni_2_P-Fe_2_P/NF electrocatalyst with a 2D nanosheet structure, using a phosphidation and acidification method (as illustrated in Figure 7d), that exhibited outstanding intrinsic performance. The morphological analysis verified the 2D structure for the ultrathin Ni-related precursors and a similar nanosheet-like structure for the Ni_2_P-Fe_2_P/NF electrocatalyst, as depicted in Figure 7e–g. The structure of the bimetallic phosphide crystal which formed the heterostructure between the Fe_2_P and Ni_2_P phases was clearly visible. The Ni_2_P-Fe_2_P/NF delivered 100 and 1000 mA cm^−2^ for the HER, requiring overpotentials of 252 and 389 mV, respectively, and operated for 48 h in a water solution of 1.0 M KOH. Furthermore, it was proven that increasing the specific surface area improved the electrocatalyst performance, since more active sites were available to enhance its efficiency. This way, not only specific surface area scattering was enhanced and thus the active sites, but also full association was permitted between the electrolyte and the active sites, thus enhancing the performance of the electrocatalyst. Furthermore, Ni-Fe helped to improve resistance to corrosion and stability, which was beneficial for the H_2_O electrolysis procedure.

### 4.3. Heterostructure

The transformation of the electrocatalyst’s structure into a heterostructure has subsequently been identified as an appealing source for the enhancement of the active sites’ density and mass transfer rate, hence enabling the improvement of the activity. Liu et al. [52] developed the CoNiP/Co_x_P/NF electrocatalysts using the chemical vapor deposition and electrodeposition methods. The NF (substrate) with a 3D structure and high conductivity was first synthesized, followed by the synthesis of CoNiP/Co_x_P/NF via chemical vapor deposition and the formation of CoNi alloy/NF through electrodeposition. The overpotential of the designed CoNiP/Co_x_P/NF electrocatalyst at 10 mA cm^−2^ was 290 mV for HER and showed stability up to 500 h in clean water. This structure had a high concentration of exposed active sites and high corrosion resistance, resulting in superior performance and long-term stability during water electrolysis. DFT-simulation findings showed that the electrocatalyst (CoNiP) possessed an appropriate thermodynamic activity for the desorption/adsorption of H_2_ [52].

### 4.4. Core–Shell Structure

Wu et al. [54] constructed a CoP_x_@FeOOH/NF electrocatalyst through a three-step process, which included hydrothermal treatment, phosphidation, and electrodeposition. CoP_x_ precursors demonstrated linear smooth NWs and heterogeneous CoP_2_ distributions. As seen from the SAED and HR-TEM images (Figure 8a–d) in the CoP_x_@FeOOH/NF electrocatalyst, the FeOOH works as the shell while the CoP_x_ works as the core in the core–shell structure.

It is widely believed that phosphide-related electrocatalysts usually exhibit excellent HER performance due to the P atoms that can easily break down H_2_ molecules and trap the H* intermediates [158,159,160]. Compared to pure phase CoP, the CoP_x_ catalyst has a relatively greater number of active sites for HER performance due to the heterogeneous phase composition (CoP-CoP_2_). This may be due to a greater number of P atoms or defective phase interfaces [161,162,163,164]. Additionally, the reported material (CoP_x_) has a nanowire mesh structure that provides beneficial features to the catalyst for H_2_ production through water electrolysis. Some of those features are mentioned below; every catalytic nanowire is exposed to react, providing a broad contact with the electrolyte molecules, and enhancing mechanical strength.

The CoP_x_ catalyst showed greater HER activity compared to other self-supported catalysts since it required low overpotential for attaining the required current densities. More precisely, 117, 190, 248, and 269 mV overpotentials were needed to attain 10, 100, 500, and 800 mA cm^−2^ current densities, respectively, with a 71.1 mV dec^−1^ Tafel slope value (Figure 8e). The catalysts which have a hydrophilic surface and nanowire mesh structure can promote H_2_ bubble release and electrolyte diffusion, leading to structural stability and outstanding catalytic durability in 1 M KOH seawater electrolyte, which was confirmed by the negligible decline in the polarization curve (Figure 8e,f) [165].

As concluded, the core–shell structures are expected to contribute to the high electrocatalyst’s performance (activity and stability) by having both corrosion-resistance and active layers, demonstrating the important role structural morphology plays. Furthermore, at an industrial-scale current density of 500 mA cm^−2^, measurements were performed to check the catalytic durability of the CoP_x_||CoP_x_@FeOOH pair through the long-term chronopotentiometric method. The results are depicted in Figure 8f, showing a potential fluctuation of only 53 mV over 80 h of continuous testing. These excellent catalytic durability results indicate that CoP_x_@FeOOH and CoP_x_ catalysts are realistic for H_2_ production by electrolysis of water [54].

## 5. Concluding Remarks

In this review article, we have discussed the production of H_2_ as an emerging energy carrier and illustrated the techniques used for this purpose. By using renewable energy sources, the electrolysis of water is the most competent technology to mitigate both energy demands and environmental pollution. Furthermore, different water electrolysis methods for H_2_ production (AWE, PEMWE, SOE, and MEC) were discussed. Additionally, electrocatalysts play a key role in H_2_ production by water electrolysis. In this regard, from the various types of electrocatalysts, only the recent developments in transition metal tellurides and transition metal phosphides have been discussed in this review, due to their promising features, such as enhanced H_2_ production rate, stability, durability, high efficiency, and low cost requirements as compared to precious commercial electrocatalysts, such as the platinum group metals (PGMs). Although the development of metal-based tellurides and phosphides is competing with PGMs, more effort is still required to enable the commercialization of water-splitting technology.

Therefore, an enhancement of the catalytic activity, as well as long-term operational stability, are highly required.

There are some points to be considered for the future research in the field.

Transition metal-based catalysts require some improvements for industrial-scale water electrolysis in terms of current density (greater than 500 mA cm^−2^) and long-term stability.TMT-based electrocatalysts have high electrical conductivity, due to their metallic character, but conductivity is not the only characteristic needed for enhancing the HER performance of the electrocatalysts. Therefore, improvements of the electronic structure of electrocatalysts are still required to change the composition; and of the structural engineering to achieve binding energy regulation of the reaction intermediates and lower the reaction energy barrier.The development of novel and advanced synthesis methods is required for large-scale production with suitable structural properties.More research is needed to explain the self-construction mechanism in the transition metal tellurides and phosphides. Furthermore, to understand the self-construction mechanisms and intrinsic properties, tests under different conditions should be performed for the self-constructions of TMT and TMP. By controlling the construction mechanism of the electrocatalysts, HER performance can be boosted.Development of simple but more accurate characterization techniques for the structural elucidation of the material may be beneficial for designing efficient electrocatalysts for HER performance.Transition metal telluride and phosphide electrocatalysts have greater electrical conductivity than other electrocatalysts due to the metallic character of transition metals.TMT and TMP electrocatalysts are better than conventional platinum-group electrocatalysts, due to their low cost.

## Figures and Tables

**Figure 1 membranes-13-00113-f001:**
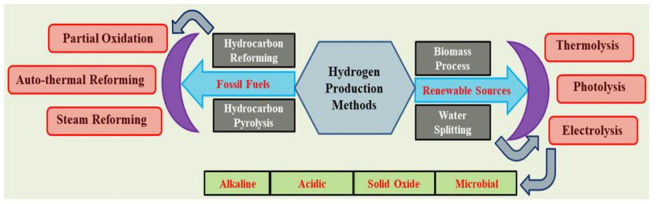
Methods for the production of hydrogen.

**Figure 2 membranes-13-00113-f002:**
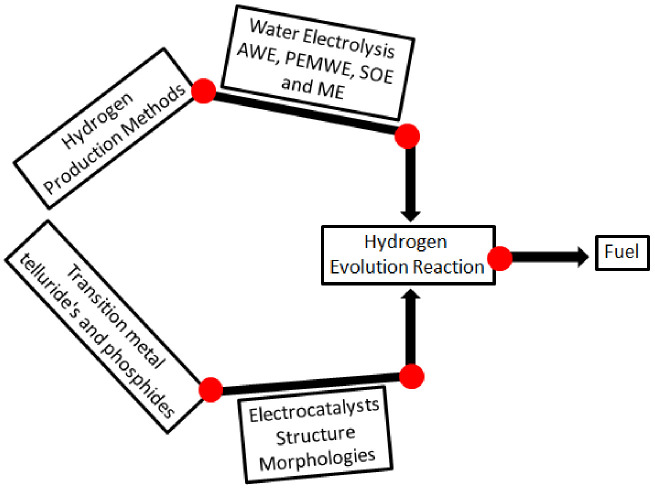
Schematic illustration of the article’s overview.

**Figure 3 membranes-13-00113-f003:**
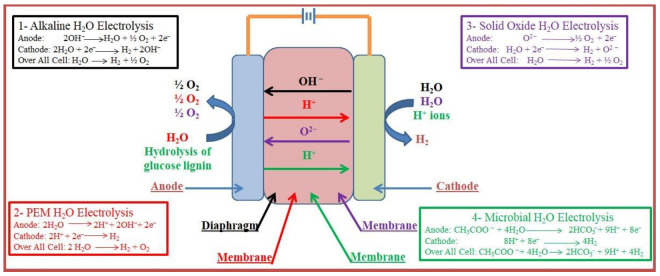
Schematic representation of the different H_2_O electrolysis techniques. Note: the different colors correspond to the four different types of H_2_O electrolysis cells: (1) black color for the alkaline electrolysis cell; (2) red color for the PEM electrolysis cell; (3) purple color for the solid oxide electrolysis cell; (4) green color for the microbial electrolysis cell.

**Figure 4 membranes-13-00113-f004:**
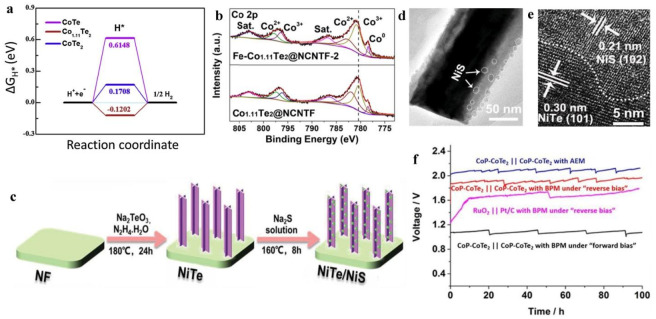
(**a**) Gibbs free energy diagram for CoTe, CoTe_2_ and Co_1.11_Te_2_ reproduced from [117]; (**b**) XPS spectra images of Fe-Co_1.11_Te_2_@NCNTF and Co_1.11_Te_2_@NCNTF, reproduced from [118]; (**c**) schematic design of the synthesis of NiTe/NiS decorated on heterojunction nanoarrays, reproduced from [119]; (**d**,**e**); transmission electron microscopy images of NiTe/NiS decorated on heterojunction nanoarrays reproduced from [119]; (**f**) catalytic stability results of BPWEM, reused from [120].

**Figure 5 membranes-13-00113-f005:**
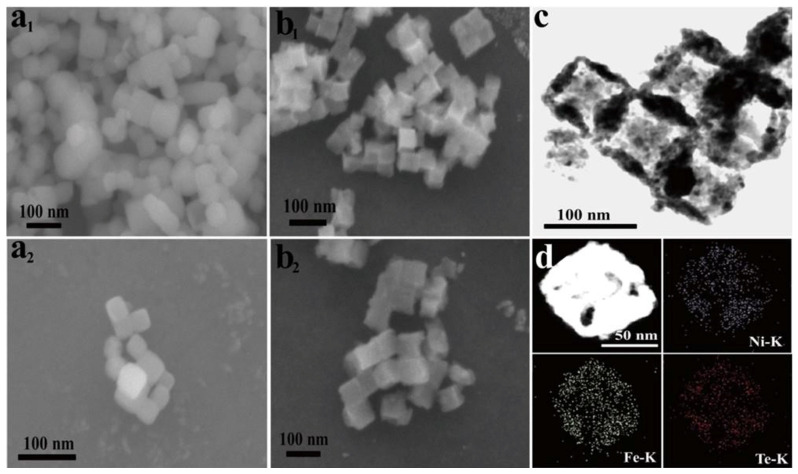
FESEM images of FeNiOOH-NCs and Te/FeNiOOH-NCs catalysts are represented in (**a_1_**,**a_2_**), (**b_1_**,**b_2_**), respectively; (**c**) TEM image of catalyst (Te/FeNiOOH-NC); (**d**) high-angle annular dark-field STEM image (Te/FeNiOOH-NC) and consistent elemental mapping images of Te, Ni, and Fe EDX. These figures were reprinted from [150].

**Figure 6 membranes-13-00113-f006:**
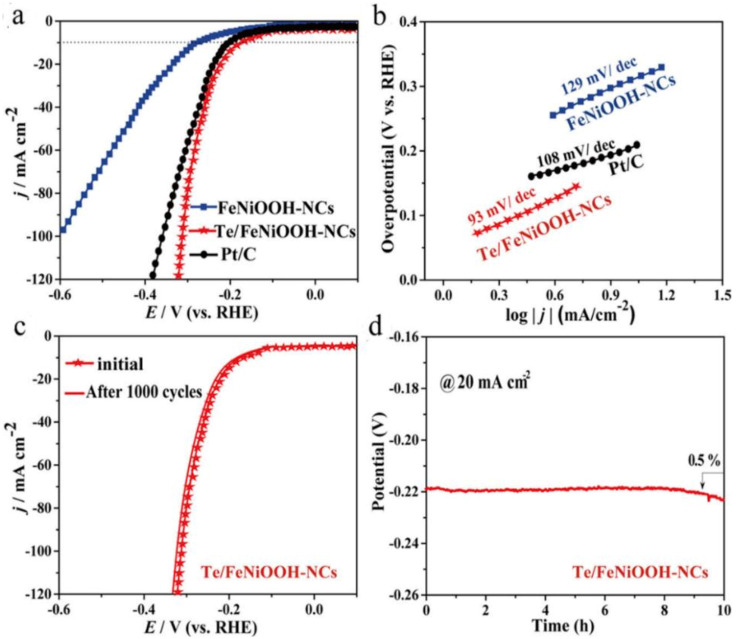
(**a**) LSV curves and (**b**) Tafel plots of Te/FeNiOOH-NCs, FeNiOOH-NCs, and Pt/C electrocatalysts for HER, measured in a 1.0 M KOH solution with 1600 rpm; (**c**) LSV curves before and after 1000 cycles of HER; (**d**) chronopotentiometry (CP) measurements for Te/FeNiOOH-NCs electrocatalyst; these figures were reprinted from [150].

**Figure 7 membranes-13-00113-f007:**
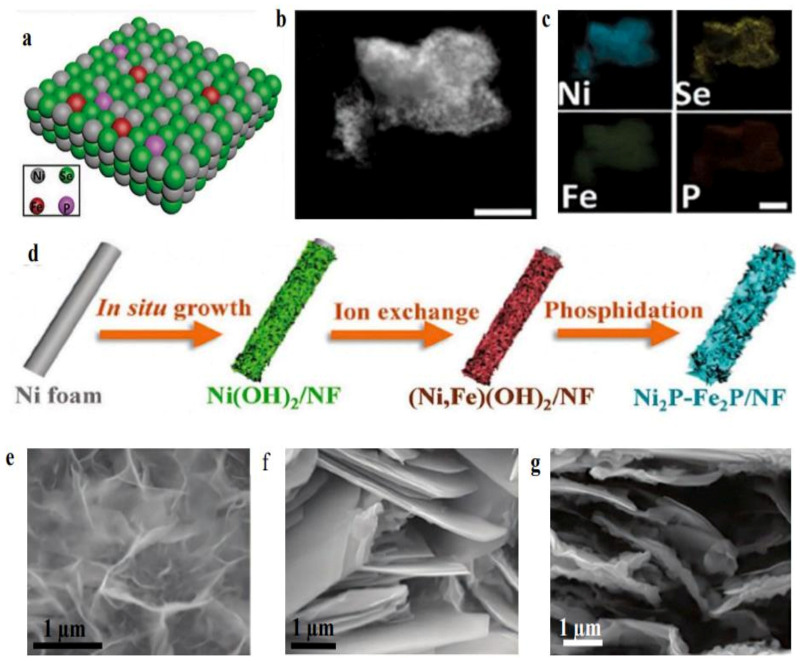
(**a**) Dual doped structure of NFs of Fe, P-NiSe_2,_; (**b**,**c**) illustrated element representations in NFs of Fe, P-NiSe_2_, reproduced from [53]; (**d**) systematic scheme for the synthesis of NFs of Fe, P-NiSe_2_; (**e**–**g**) are the SEM pictures of electrocatalysts Ni(OH)_2_/NF, Ni_2_P-Ni_5_P_4_/NF, and Ni_2_P-Fe_2_P/NF, correspondingly, reprinted from [55].

**Figure 8 membranes-13-00113-f008:**
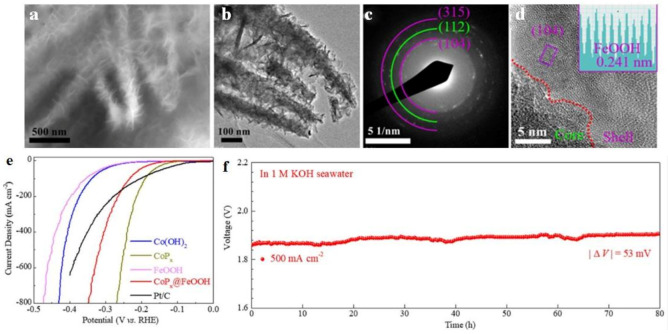
(**a**–**d**) SEM, TEM, HRTEM images and SAED pattern of CoP_x_@FeOOH; (**e**) HER polarization curves of various electrocatalysts; (**f**) chrono-potentiometric measurement of CoP_x_||CoP_x_@FeOOH in 1.0 M KOH water electrolyte, reused from [54].

**Table 1 membranes-13-00113-t001:** Performance of TMP- and TMT-based electrocatalysts for hydrogen evolution reaction (HER).

Sr. No	Electrocatalysts	HER Medium	Current Density (mA cm^−2^)	Overpotential (mV)	Tafel Slope (mV dec^−1^)	Refs
1	Co_2_P@Cu nanostructure	Alkaline	10 100	99.7 303.2	48.8	[56]
2	Co_2_P/Ni_2_P/CNT	Acidic Alkaline	10 10	151 202	41.64 ----	[57]
3	FeNiP/MoO_x_/NiMoO_4_/NF	Alkaline	10 100	16 97	21.2	[58]
4	Ni_2_P@NC/NF	Alkaline	10	93	----	[59]
5	MoP-Ru_2_P/NPC	Alkaline	10 10	47 82	36.93 64.99	[60]
6	CoTe_2_/Ti_3_C_2_T_x_	Alkaline	10	200	95	[61]
7	NiFe_2_O_4_/NiTe	Alkaline	10	148.8	73.67	[62]
8	NiTe-HfTe_2_/g-C_3_N_4_	Alkaline	10	71	75	[63]
9	Co,Ni-MoTe_2_	Acidic	10	−82	----	[64]
10	CoP/Ni_2_P@Co(OH)_2_	Acidic Alkaline	10 10	68 39	68 55	[65]

(NF = Ni foam; CNT = carbon nanotubes; NPC = porous N and P co-doped carbon; g-C_3_N_4_ = graphitic carbon nitride).

## Data Availability

No new data were created or analyzed in this study. Data sharing is not applicable to this article.

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
