# Peer review of "Recent Advances in Transition Metal Tellurides (TMTs) and Phosphides (TMPs) for Hydrogen Evolution Electrocatalysis"

_membranes, 2023, doi:10.3390/membranes13010113_

Round 1

Reviewer 1 Report

The short review focuses on recent half-decade progress in the electrocatalytic performance of transition metal tellurides (TMTs) and phosphides (TMPs) towards highly efficient production of H2. Overall, the whole manuscript is well organized. But some points should be addressed, as follows:

1. The advantages of TMTs and TMPs for OER should be highlighted rather than just low cost and abundance.

2. A schematic diagram should be needed to summarize the idea of the manuscript.

3. In the two parts of “TMTs and TMPs-based electrocatalysts for HER”, different subheadings should be provided according to the improvement strategies for enhance the catalytic activity.

4. The perspectives also should be discussed in detail.

5. Please note the units of physical quantities, such as electricity or heat et al.

Author Response

REVIEWER 1

The short review focuses on recent half-decade progress in the electrocatalytic performance of transition metal tellurides (TMTs) and phosphides (TMPs) towards highly efficient production of H2. Overall, the whole manuscript is well organized. But some points should be addressed, as follows:

  1. The advantages of TMTs and TMPs for HER should be highlighted rather than just low cost and abundance.

Answer: As requested the advantages of TMTs and TMPs has been added in the manuscript.

(see P.2-3 the highlighted text: ’Between these attractive catalysts, transition-metal-tellurides (TMTs) and transition-metal-phosphides (TMPs) attracted much attention in recent years….components of the material, resulting in catalytic performance enhancement [43-49].’)

  1. A schematic diagram should be needed to summarize the idea of the

Answer: According to reviewer’s suggestion, a schematic diagram that explains the idea of this review has been added (see Figure 2. Schematic illustration of the article’s overview).

  1. In the two parts of “TMTs and TMPs-based electrocatalysts for HER”, different subheadings should be provided according to the improvement strategies for enhance the catalytic activity.

Answer: Subheadings related to TMTs and TMPs-based electrocatalysts have been added to the manuscript. More precisely, chapter 3 pertains to TMTs, while chapter 4 pertains to TMPs.

  1. The perspectives also should be discussed in detail.

Answer: Perspectives added in the manuscript.

  1. Please note the units of physical quantities, such as electricity or heat et al.

Answer: We checked the units of physical quantities.

Reviewer 2 Report

A. There are obvious format problems in the upper and lower markers in the article, such as lines 121 and 122 in 2.1, the chemical formulas of water and oxygen in Figure 2, and line 5 in 2.3.There are obvious errors that need to major revision by the author.

B. Since the equation 1 in the article is an equation, why there is no equivalent relationship needs to be carefully revised by the author.

C. The English writing of this manuscript is rather poor. There are many mistakes in the whole manuscript. For example, in the introductionRen et al. prepared Ni2P-Fe2P catalysts that showed outstanding stability and activity, and CoPx@FeOOH catalysts that exhibited 89 good stability for 80 h at 500 mA cm−2 current density and an overpotential of 283 mV at 100 mA cm−2 in 1.0 M KOH-containing water. I only listed one here. The author should carefully revise and polish the manuscript.

D. This article is a review of transition metal tellurides and transition metal phosphides. However, there are no specific examples of transition metal tellurides in electrocatalytic applications in the introduction, which leads to the lack of convincing summary in the last paragraph of the introduction. Here, the author needs to add the corresponding content.

E. Most of the figures have very low resolution. They deserve a little more effort in presentation (e.g. Figure 6a-d).

F. For the content of Figure 4, the author should not only provide electrochemical performance, should reflect the structure of the composite material on the electrocatalytic performance improvement.

G. The author should explain in the concluding section the novelty of the review in comparison to many other review articles on the same topic and add some perspective.

Author Response

REVIEWER #2

  1. There are obvious format problems in the upper and lower markers in the article, such as lines 121 and 122 in 2.1, the chemical formulas of water and oxygen in Figure 2, and line 5 in 2.3. There are obvious errors that need to major revision by the author.

Answer: The indicated errors have been removed from the revised manuscript. Overall the manuscript was thoroughly checked for carelessness mistakes. 

  1. Since the equation 1 in the article is an equation, why there is no equivalent relationship needs to be carefully revised by the author.

Answer: Equation 1 has been revised, as follow:

2H2O + [Electricity 237.2 kJ/mol + Heat 48.6 kJ/mol]           à 2H2 + O2     (1)

  1. The English writing of this manuscript is rather poor. There are many mistakes in the whole manuscript. For example, in the introduction “Ren et al. prepared Ni2P-Fe2P catalysts that showed outstanding stability and activity, and CoPx@FeOOH catalysts that exhibited 89 good stability for 80 h at 500 mA cm−2 current density and an overpotential of 283 mV at 100 mA cm−2 in 1.0 M KOH-containing water. I only listed one here. The author should carefully revise and polish the manuscript.

Answer: We have carefully checked the entire manuscript and made the appropriate improvement.

  1. This article is a review of transition metal tellurides and transition metal phosphides. However, there are no specific examples of transition metal tellurides in electrocatalytic applications in the introduction, which leads to the lack of convincing summary in the last paragraph of the introduction. Here, the author needs to add the corresponding content.

Answer: Thanks for highlighting deficiency in the introduction section. This problem has been solved by adding the corresponding content.

(see P.2-3 the highlighted text: ’Between these attractive catalysts, transition-metal-tellurides (TMTs) and transition-metal-phosphides (TMPs) attracted…. resulting in catalytic performance enhancement [43-49].’ and ‘Anyhow, there are many examples of…. CoP/Ni2P@Co(OH)2.’)

  1. Most of the figures have very low resolution. They deserve a little more effort in presentation (e.g. Figure 6a-d).

Answer: Resolution of the images has been improved.

  1. For the content of Figure 4, the author should not only provide electrochemical performance, should reflect the structure of the composite material on the electrocatalytic performance improvement.

Answer: According to the suggestion of the reviewer, reflection of structure of composite material on the electrocatalytic performance improvement has been discussed in the manuscript.

(see P.9-10 the highlighted text: ‘The synthesized catalysts Te/FeNiOOH-NCs and FeNiOOH-NCs were characterized through SEM… and Ni elements (Figure 5d).’)

  1. The author should explain in the concluding section the novelty of the review in comparison to many other review articles on the same topic and add some perspective.

Answer: We included in the conclusion section the novelties of the present review article. (see the highlighted text on the chapter: concluding remarks)

Reviewer 3 Report

1. Authors mention that H2 energy attained the environmentally friendly method. Please explain how it is environmentally friendly. You may refer following references: Chemical Communications, 2020, 56, 6953-6956.

2.     Authors mention alkaline electrolytes used for HER, except acidic electrolytes. Give suitable examples of as mentioned materials for HER in acidic electrolytes.  

3.     Authors mention many materials for HER without performance. Please give a proper explanation with a table form.

4.     Authors mention that Pt/C shows low activity for HER in both the LSV curve and Tafel slope compared to Te/FeNiOOH-NC. Please give the appropriate information about Pt/C (percentage of Pt and C) to show less activity for HER.

5.     Fig shows the chronopotentiometry (CP) measurements of  Te/FeNiOOH-NCs at 20 mA cm−2 fig. 4d shows that potential is stable up to 8 h but after that suddenly increases. Please explain why it suddenly increased.

6.     HER polarization curve of the CoPx electrocatalyst in Fig. 6 e shows excellent activity compared to Pt/C. Please mention the percentage of Pt and C for comparison with it. 

7.     CoPx material is used as an excellent material for HER, as shown in Fig. 6 e but different material such as CoPx||CoPx@FeOOH is used for stability in fig 6f. Please give the proper explanation why the use of different materials.   

8.     HER polarization curves of various electrocatalysts in Fig. 6 e and stability of materials in Fig. 6 f show different potential windows.  Please give the proper explanation for that.

Author Response

REVIEWER #3

REVIEWER #3

  1. Authors mention that H2 energy attained the environmentally friendly method. Please explain how it is environmentally friendly. You may refer following references: Chemical Communications, 2020, 56, 6953-6956.

Answer: Thanks for sharing valuable reference source. It has been added in the revised manuscript with the required explanation.

  1. Authors mention alkaline electrolytes used for HER, except acidic electrolytes. Give suitable examples of as mentioned materials for HER in acidic electrolytes.

Answer: Thanks for highlighting this important point “acidic electrolytes”, but in the current review our focus is on alkaline electrolytes.

  1. Authors mention many materials for HER without performance. Please give a proper explanation with a table form.

     Answer: According to reviewer’s suggestion, HER results in the form of table has been added in the revised manuscript.

  1. Authors mention that Pt/C shows low activity for HER in both the LSV curve and Tafel slope compared to Te/FeNiOOH-NC. Please give the appropriate information about Pt/C (percentage of Pt and C) to show less activity for HER

Answer: We thanks very much for the reviewer’s careful evaluation of the manuscript. The percentage of Pt in Pt/C catalyst was added in the revised manuscript.

  1. Fig shows the chronopotentiometry (CP) measurements of Te/FeNiOOH-NCs at 20 mA cm−2 4d shows that potential is stable up to 8 h but after that suddenly increases. Please explain why it suddenly increased.

Answer: An attempt to explain the 0.5% loss of potential according to the literature has been done in the revised manuscript.

(see P.11 the highlighted text: ‘Post-characterizations (XRD and FESEM images…. providing a more profound knowledge of the catalyst's post-HER morphology.’)

  1. HER polarization curve of the CoPx electrocatalyst in Fig. 6e shows excellent activity compared to Pt/C. Please mention the percentage of Pt and C for comparison with it. 

Answer: As requested the percentage of Pt in Pt/C catalyst was added in the revised manuscript.

  1. CoPx material is used as an excellent material for HER, as shown in 6 e but different material such as CoPx||CoPx@FeOOH is used for stability in fig 6f. Please give the proper explanation why the use of different materials.  

Answer: We added the requested explanation in the revised manuscript. 

(see P:13-14 the highlighted text: ‘It is widely believed that phosphide-related electrocatalysts usually exhibit excellent HER performance due to… for H2 production by electrolysis of water [163].’)

  1. HER polarization curves of various electrocatalysts in 6e and stability of materials in Fig. 6f show different potential windows.  Please give the proper explanation for that.

Answer: According to reviewer’s suggestion, this detail has been added in the manuscript.

Round 2

Reviewer 3 Report

Revision is satisfactory manuscript can be accepted for publication.

Author Response

Reply to Academic Editor’s Comments

Dear Editor,

Thank you very much for providing us another opportunity to further improve the article. We have revised the article according to comments of reviewers/editor and changes in the marked up revised manuscript are highlighted in yellow.

Academic Editor’s Comments

The responses to the Reviewer 2 are not sufficient to resolve whole issues raised. Thus, Figure 2 and concluding remarks should be carefully revised again.

-Figure 2 should be sophisticatedly made based on Section 3 and 4 in the revised manuscript.

Answer 1: Figure 2 has been reviesed to better reflect article’s overview.

-Using Table in Concluding Remark section is not appropriate. Table 1 should be in the content section. The concluding remarks should highlight the reason why the transition metal tellurides (TMTs) and phosphides (TMPs) are of importance in water electrolysis technology compared to the other electrocatalysts.

Answer 2: Table 1 has been shifted and excluded from the conclusion portion. Moreover, few more points have been added in the conclusion remarks for better highlight the why TNTs and TMPs are of importance in water electrolysis technology compared to the other electrocatalysts.
